# Current State and Future Directions in the Therapy of ALS

**DOI:** 10.3390/cells12111523

**Published:** 2023-05-31

**Authors:** Laura Tzeplaeff, Sibylle Wilfling, Maria Viktoria Requardt, Meret Herdick

**Affiliations:** 1Department of Neurology, Rechts der Isar Hospital, Technical University of Munich, 81675 München, Germany; 2Department of Neurology, University of Regensburg, 93053 Regensburg, Germany; sibylle.wilfling@klinik.uni-regensburg.de; 3Center for Human Genetics Regensburg, 93059 Regensburg, Germany; 4Formerly: Department of Neurology with Institute of Translational Neurology, Münster University Hospital (UKM), 48149 Münster, Germany; m.requardt@uni-muenster.de; 5Precision Neurology, University of Lübeck, 23562 Luebeck, Germany

**Keywords:** amyotrophic lateral sclerosis, ALS, motor neuron disease, MND, medication, therapy, supportive therapy, clinical trials, personalized medicine

## Abstract

Amyotrophic lateral sclerosis (ALS) is a rapidly progressive neurodegenerative disorder affecting upper and lower motor neurons, with death resulting mainly from respiratory failure three to five years after symptom onset. As the exact underlying causative pathological pathway is unclear and potentially diverse, finding a suitable therapy to slow down or possibly stop disease progression remains challenging. Varying by country Riluzole, Edaravone, and Sodium phenylbutyrate/Taurursodiol are the only drugs currently approved in ALS treatment for their moderate effect on disease progression. Even though curative treatment options, able to prevent or stop disease progression, are still unknown, recent breakthroughs, especially in the field of targeting genetic disease forms, raise hope for improved care and therapy for ALS patients. In this review, we aim to summarize the current state of ALS therapy, including medication as well as supportive therapy, and discuss the ongoing developments and prospects in the field. Furthermore, we highlight the rationale behind the intense research on biomarkers and genetic testing as a feasible way to improve the classification of ALS patients towards personalized medicine.

## 1. Introduction

Amyotrophic lateral sclerosis (ALS) is a rapidly progressive neurodegenerative disorder, characterized by the loss of upper motor neurons in the motor cortex, and lower motor neurons in the brainstem and spinal cord, ultimately impacting respiratory function and resulting in respiratory failure of approximately three to five years after symptom onset. Additional overlapping symptoms, such as cognitive and behavioral changes, or even frontotemporal dementia, may occur in ALS patients, leading to a very heterogenous clinical picture [1,2].

Currently, three pharmaceutical compounds with an effect on disease progression are approved and differ by country: the glutamate antagonist Riluzole (orally available in different forms: tablet, film, or liquid), the antioxidant Edaravone, and the recently introduced Sodium phenylbutyrate/Taurursodiol [3,4,5]. By slowing down disease progression (measured by the revised ALS Functional Rating Scale, ALSFRS-R) [6], these drugs can prolong autonomy and increase survival by a few months [7,8,9]. A fourth agent, a combination of dextromethorphan hydrobromide and quinidine sulfate, is approved for the symptomatic treatment of frontal disinhibition. Oxidative stress, excitotoxicity, mitochondrial and proteasomal dysfunctions, RNA metabolism, altered synaptic function, disturbed axonal transport, and neuroinflammation are, among other pathways, considered as important contributors to ALS disease development [10,11]. However, oxidative stress and excitotoxicity are the only two pathways targeted by the current FDA-approved drugs. Therefore, targeting other disease mechanisms may be key to the identification of new therapeutic drugs that could allow for additive or more effective treatment of ALS. Clinical trials attempt to address several dysregulated pathways but the transition from a mouse model to human clinical treatment remains challenging and there are questions as to whether current methods require a general modification [12,13,14]. While many current studies are designed to discover new drugs, others focus on the improvement of drug administration and efficacy [15]. Few do, however, focus on supportive and palliative care. In order to maintain functionality and autonomy, physical and speech therapy, occupational therapy, and robotic help systems are only some ways to maintain functionality and, therefore, patients’ autonomy as long as possible, increasing the perceived quality of life [16,17,18]. Nutritional therapy can help maintain body weight [19,20] and respiratory failure with the resulting hypercapnia can be treated by noninvasive or invasive ventilation, a method that can prolong survival [21]. Addressing supportive and palliative care not only eases symptoms caused by the disease but may also actively influence the individual disease course.

The estimated prevalence of ALS in 22 European countries was 121,028 cases in 2020 and the global incidence is reported to be 1–2.6:100,000 [22,23]. In the past years, many studies have focused on ALS as a disease, and the gained knowledge is immense. However, why is it still so complicated to find a suitable treatment able to stop ALS progression once and for all? In addition, what are the directions clinicians and researchers should observe when focusing on the improvement of current medication and therapeutic options? We will discuss the ongoing developments and future perspectives regarding drug improvement as well as technological innovation and general modification in study designs.

A challenge in the treatment of early-stage ALS remains the discrepancy between disease onset and diagnosis, differing by about nine to twelve months on average [24,25]. In addition, some phenotypic manifestations do not fulfill diagnostic criteria in early disease stages and ALS mimics also blend into cohorts and complicate the process of diagnosis finding. Delayed diagnosis makes it difficult to treat ALS patients during early disease stages but initiation of therapy at symptom onset or even in presymptomatic patients could help slow down the fundamental neurodegenerative process. Many altered pathways and genetic mutations have been discovered and associated with ALS which makes it, up to date, impossible to find “the one” cause for the disease, and the one therapy to suit all [15,26]. Arising discussions often address the question of whether “the” one ALS, as currently defined, might in fact not be a group of different diseases with overlapping symptoms but diverging pathological mechanisms. If so, would these differences not revolutionize our clinical-trial culture? How could future trial groups be stratified? By genetic, clinical, or even molecular-based subgroups? Genetic testing and suitable blood- or spinal-fluid-based biomarkers might thus be the future way to personalize ALS therapy. Can these stratifications ultimately lead to a more personalized, more precise, and more promising search for disease curation?

In this review, we summarize the current state of ALS therapy. We extensively discuss medical as well as supportive therapy options as they are both important aspects of ALS disease management and, in our opinion, inseparable in the current state of ALS treatment. We further discuss the ongoing clinical trials with a particular focus on small molecules, but also clinical trials focusing on gene-specific therapy (antisense oligonucleotides and viral vectors delivering RNA interference and CRISPR/Cas9) as well as monoclonal antibody and stem-cell therapy. Different ALS mechanisms that are currently not targeted in clinical trials are also addressed. Lastly, we highlight the current difficulties of validating new drugs in clinical trials and suggest the stratification of ALS patients considering clinical characterizations, genetics, and current studies on ALS-linked biomarkers as ways to create a more personalized ALS treatment.

## 2. Current Treatment Options

### 2.1. Medical Treatment

#### 2.1.1. Riluzole

For a long time, Riluzole (6-(trifluoromethoxy)-2-aminobenzothiazole) was the only treatment option available for patients with ALS. The targeted pathway was mainly aimed at the reduction of excitotoxicity, specifically the excitotoxicity of glutamate. The positive effect of Riluzole is thought to result from different actions. Firstly, its action as a sodium channel blocker on presynaptic neurons decreases glutamate release into the synaptic cleft. Furthermore, glutamate reuptake is increased through the activation of astrocytic excitatory amino acid transporter 2 (EAAT2) channels and, additionally, aminomethylphosphonic acid (AMPA) and N-methyl-D-aspartate (NMDA) glutamate receptors on postsynaptic neurons are inhibited noncompetitively. Lastly, gamma-aminobutyric acid (GABA) reuptake seems to be reduced and GABA receptors potentiated [27,28,29,30,31].

Riluzole was approved by the FDA in 1995 and subsequently the EMA in 1996. In the first study examining the treatment of ALS patients with Riluzole, 155 participants were included in a prospective, double-blind, placebo-controlled trial. The intervention group received a daily dose of 100 mg of Riluzole, the standard dosage still valid today. After a treatment period of one year, a significant delay of disease progression was shown [4]. A further multicenter double-blind and placebo-controlled study with 959 participants confirmed these results in 1996 [32], suggesting a survival benefit of two to three months in both studies. However, real-world analyses suggest a higher survival benefit, ranging up to 19 months [33]. Most importantly, the study was able to identify Riluzole as a generally well-tolerated drug and determined the side effects, including asthenia, nausea, gastrointestinal problems, as well as augmentation in hepatic enzymes.

The Riluzole drug was first developed as an oral tablet of 50 mg taken twice a day but given the fact that many ALS patients are struggling with dysphagia, the development of a bioequivalent oral suspension of Riluzole (5 mg/mL) was essential [34,35]. Furthermore, recently, an oral-film formulation form of Riluzole has been developed that poses another option to simplify administration and ensure the continuation of medical treatment [36,37].

#### 2.1.2. Edaravone

After the approval of Riluzole, no new drug entered the market for years until, in 2015, Edaravone (3-methyl-1-phenyl-2-pyrazoline-5-one) was approved in Japan and South Korea. After its FDA approval in 2017, China’s NMPA and Switzerland followed with admissions in 2019, Indonesia in 2020, and, finally, Malaysia and Thailand in 2021. Edaravone has been used in acute-phase stroke intervention in Japan for years [38] and was designed to reduce oxidative stress and neuroinflammatory response by scavenging free radicals [39,40,41]. Its use in ALS treatment is based on a randomized, double-blind, placebo-controlled phase III trial with 137 participants [42]. Inclusion criteria allowed only participants with early stage ALS defined as disease duration of two years or less, ALSFRS-R scores of at least two for every item of the ALSFRS-R, forced vital capacity (FVC) of 80% or more, as well as a probable or definite ALS diagnosis as defined by the revised El Escorial criteria [42]. Prior to that, a previous phase III trial had not shown any significant overall effect of this drug [5]. Significant treatment effects were only shown after post hoc subgroup analysis with the above-mentioned criteria for early disease, which were then used in the second phase III study [43]. Results from the second phase III trial were a 33% reduction of ALSFRS-R decline over the 24-week treatment period and significantly better scoring on the ALSAQ-40 (forty item ALS assessment questionnaire; quality of life assessment tool). However, further studies did not yield significant effects, again [44,45]. Typical side effects of Edaravone treatment were bruising, gait disturbances, headaches, and skin conditions [5,46,47].

Up to date, Edavarone is applied intravenously at a dosage of 60 mg/day. While the first administration takes place over a period of 14 days, all following monthly cycles take place for ten days. However, the convenience of this therapy method is below the one of orally assumable medicine. In 2022, an oral suspension of 105 mg/day was approved by the FDA. Simultaneously, there is a phase III trial conducted by Ferrer (TRICALS) for evaluating the safety and potential efficacy of FNP122, another oral application form of Edaravone.

#### 2.1.3. Sodium Phenylbutyrate and Taurursodiol

Most recently, Sodium Phenylbutyrate and Taurursodiol (also known as PB-TURSO or PB-TUDCA [tauroursodeoxycholic acid] (Sodium 4-pheylbutanoate and 2-[(3α, 7β-dihydroxy-24-oxo-5β-cholan-24-yl) amino] ethane sulfonicacid, dihydrate) have been released as a fixed-dose combination. This drug is currently approved by the FDA and in Canada and both substances were previously approved for medical use separately. While Sodium Phenylbutyrate can be used in the treatment of hyperammonemia due to urea-cycle disorders [48], Taurursodiol has its place in the treatment of chronic cholestatic liver disorders [49]. Both have been identified as inhibitors of neuronal apoptosis and their combined action is proposed to reduce neuronal cell death and oxidative stress by decreasing stress in the endoplasmic reticulum (ER) and mitochondrial dysfunction [50,51,52,53]. FDA approval was then based on results from a phase II multicenter, randomized, double-blinded trial (CENTAUR) that tested the safety and efficacy of a fixed-dose combination in 137 participants [3,54,55]. The combination of both substances slowed down disease progression (measured by the ALSFRS-R) by about 25% compared to the placebo group, and initiation of PB-TURSO treatment at baseline resulted in prolonged median survival of about 6.5 months. A phase III trial for the fixed-dose combination is currently ongoing and the recruitment of 664 participants was completed in February 2023 (NCT05021536). The most common adverse events during the phase II trial were diarrhea, abdominal pain, nausea, and upper-respiratory-tract infections. Gastrointestinal-related adverse reactions occurred more frequently during the first three weeks of treatment [3].

Sodium phenylbutyrate/Taurursodiol was designed as an oral suspension and produced as a powder mixed with room-temperature water. The correct dosage contains 3 g of sodium phenylbutyrate and 1 g of Taurursodiol; it is initially administered once a day for three weeks and then twice a day.

### 2.2. Supportive Therapy

Supportive and palliative care is highly important to patients but rarely focused on in research. Nonmotor symptoms involving neuropsychiatric, autonomic, gastrointestinal, and vascular systems affect between 5% and 80% of people with ALS [56]. Thus, symptomatic treatment forms an important part of therapy in ALS patients and must be adapted to the clinical course. This includes symptomatic medical treatment for certain symptoms such as mood disturbances, pain, and spasticity that are usually not specially approved for ALS (Figure 1a). The combination of dextromethorphan hydrobromide (20 mg) and quinidine sulfate (10 mg) was FDA-approved for ALS in 2011 for symptoms of frontal disinhibition such as frequent, involuntary, and often sudden episodes of crying and/or laughing [57]. While physical, occupational, and speech therapy as well as robotic assistance can maintain patients’ autonomy, nutrition and home ventilation therapy aim to conserve somatic wellbeing. Studies have shown that access to and care through a tertiary ALS center can be a positive predictor [58] and are associated with increased six-month survival as well as reduced time to noninvasive ventilation (NIV) and gastrostomy [59]. Counseling and palliative care need to be discussed (Figure 1b). Access to palliative care can improve the quality of life of patients and primary caregivers [60]. End-of-life discussions are an important part of patient care, especially as there is evidence suggesting that these discussions are often conducted with delay [61,62]. Psychotherapy can alleviate depression and anxiety as studies on palliative care suggest and should therefore be offered, but there are only a few ALS-specific studies on this topic [63,64]. Physical and speech therapy, occupational therapy, and robotic help systems are only some ways to help maintain functionality and, therefore, patients’ autonomy for as long as possible, increasing the perceived quality of life [16,17,18]. Meanwhile, nutritional therapy may help in maintaining body weight and avoiding excessive weight loss by changing the quality and quantity of calorie intake as well as the administration method [19,20]. During the final disease phase, progressive respiratory failure is addressed by the implementation of noninvasive and invasive ventilation attempting to alleviate hypercapnic symptoms and, in some cases, prolong survival [21]. Until a satisfactory causative treatment is found, supportive therapy and efforts to improve symptomatic medication remain of the utmost importance during disease accompaniment. 

#### 2.2.1. Mobility

##### Physical and Occupational Therapy

An important part of supportive treatment for ALS patients is active and passive physical therapy (Figure 1c). Moderate active physical therapy has been shown to have a positive impact on the stabilization and maintenance of physical activity and ALSFRS-R levels when compared to passive therapy [16,65]. A frequently prescribed therapy in ALS patients, a combination of stretching and motion exercises, but also endurance and aerobic movement therapy, is considered to be safe to perform [66]. While no particular training has proven to be superior regarding results in functionality, respirational capacity, or overall survival, a combination of all of the above can have a positive effect on the perceived quality of life and reduce subjective fatigue [67].

As an extension of physiotherapy, positive psychological and physical effects have been reported for motor-assisted movement exercisers (MME). These devices are for patients with muscle stiffness or weakness and can be used for physical exercise in the domestic area. Reports have shown positive psychological and physical effects [68] and more than 50% of frequent MME users report increased general wellbeing and reduced feelings of muscle stiffness.

Occupational therapy generally focuses on daily function while assessing, for example, the necessity for assistive devices and developing status-adapted strategies for tasks of everyday life and energy conversion [69,70,71]. The benefit of occupational therapy in ALS has been shown in small-sample studies. Music therapy, for example, led to an improved quality of life [72] and a single-subject study showed marked improvements in quality of life when involving aquatic therapy [73].

##### Robotics

Developments of robotic assistance aim, among other things, to battle progressive mobility restrictions experienced during ALS patients’ individual disease course (Figure 1c). A questionnaire-based study found more than 70% of ALS patients to be in favor of establishing robotic technology [68]. The prospect of greater independence increased the number of patients interested in robotic assistance for simple tasks such as serving drinks or handling an electronic device patients preferred robotic over human assistance [68,74]. Currently, there are robotic solutions supporting patient independence and autonomy in different areas. 

Addressing decreased mobility of the upper limbs, there are single-task devices specialized in performing, for example, food intake (e.g., “My Spoon” or “Obi”) [74]. Furthermore, robotic lightweight arms, either fixed to a stationary workstation or mounted to a wheelchair, allow more complex movements within a certain radius [75]. Beyond these, there are efforts to develop wearable robotics, assisting the patient’s body movement by exoskeletal support. A recent study worked on a lightweight wearable robotic device for assisted shoulder movement of patients with ALS, in which the device was able to compensate for continuing physical deterioration in 2 of 10 evaluated participants for a period of six months [76].

Exoskeletal approaches to compensate for lower-limb weakness and the related immobility are, for now, not focused on in ALS cohorts. Based on a recent review of wearable robotic exoskeletons in patients with spinal cord injuries, there are remarkable technological challenges that might impede effective everyday use in ALS patients. For example, the mean wearing time of different exoskeletons was around 10 min, and data regarding long-term use beyond 24 weeks is missing [77]. Generally, control of exoskeleton robotic systems can be realized through position, force, and speed sensors, or by using electromyographic (EMG) or electroencephalographic (EEG) signals [78]. Electrical wheelchairs are established tools for mobility in patients with different neurological diseases and are known to increase the quality of life and comfort while decreasing pain in ALS patients [79,80]. Addressing the topic of steering without the necessity of precise fine motor skills, electromyographic surface sensors applied to temporalis muscles as well as autonomous navigation have been tested successfully [81,82,83]. Furthermore, electric wheelchairs controlled via eye-tracking devices are being investigated and there are already commercially available products on the market [84,85,86,87]. In patients with severe disability and limited motor function, electrically adjustable beds can help with changing posture, especially when coupled with an eye-tracking system such as the “Environment Control in Amyotrophic Lateral Sclerosis (ECO-ALS) System” [88]. 

The field of robotics is also highly interesting regarding the topic of bulbar symptoms. For patients with the beginning of impairment of speech and preserved ability to use a touchscreen communication device, early access has been shown to positively influence the quality of life and may be especially helpful in improving communication skills in advanced disease stages [89]. An increased quality of life can be observed in severely dys- or anarthric patients when enabled access to eye-tracking devices for communication purposes [17]. During the last years, research has further focused on brain–computer interfaces which work independently from the residual motor activity and record neuroelectric signals in the brain [90]. Though not in standard clinical use yet, the first successful demonstrations of home use have been described, mainly for communication [91,92,93,94]. 

There are constant efforts for technological improvement in order to help preserve patients’ independence; however, there are few studies objectifying the benefits and possible disadvantages of robotics. We found no study or clinical trial regarding a possible positive effect on disease progression and survival, even though one could hypothesize, that patients with access to robotics, for instance by using an electrical wheelchair with resulting passive joint movement, may have a survival benefit when compared to advanced-stage patients with impaired mobility or bed-bound patients. This fact should be taken into consideration for future studies, as technological improvement might help in maintaining autonomy.

#### 2.2.2. Speech and Swallowing

##### Speech Therapy and Communication Devices

Most patients develop bulbar symptoms such as dysarthria and dysphagia during their individual disease course [95]. Speech therapy is not only a valuable therapeutic component of disease management but can be an effective method to screen for dysphagia as well (Figure 1d) [95,96,97]. Therapists can help by teaching adaptive eating strategies and adjustment to communication devices in advanced dysarthria or anarthria. Unfortunately, studies evaluating the benefit of speech therapy in ALS, and progressively dysarthric patients in general, are rare. A British survey further elucidated the lack of validated treatment guidelines for speech therapists in progressive dysarthria [98]. The speech therapy’s effect alone is smaller than the effects and benefits of communication devices themselves when looking at the quality of life in late dysarthria stages, indicating the importance of high-technology communication devices as a source for quality of life [99].

##### Nutritional Therapy

Numerous publications have discussed nutritional therapy and its impact on overall survival in ALS patients. While antioxidant or anti-inflammatory nutrients might be beneficial in nutritional therapy [100], most studies focus on weight or body mass index (BMI) loss as an adverse prognostic factor [101,102]. Patients with ALS often show a decrease in body mass years before symptom onset [20] and weight loss from disease onset to diagnosis is associated with a reduced overall survival [103,104]. Not only do patients struggle with reduced oral caloric intake due to dysphagia but are frequently affected by hypermetabolism, intestinal problems due to medication intake, and loss of appetite [105,106]. Thus, malnutrition and underweight can be caused by a combination of several factors, all leading to higher morbidity and mortality.

While patients presenting a bulbar phenotype are more often affected by dysphagia and, therefore, malnutrition and underweight, it has been observed that patients with a rapidly progressing spinal phenotype, but no signs of dysphagia, can suffer severe weight loss, a similarly poor prognosis and early need for ventilation [103]. Again, this fact highlights the complexity of weight decline in motor neuron diseases.

The ESPEN guidelines for neurological diseases, therefore, suggest a 3-monthly follow-up for nutritional status, BMI calculation and dysphagia screening [107]. Depending on the baseline BMI, weight gain (when BMI is <25 kg/m^2^), maintenance of body weight (when BMI is between 25–35 kg/m^2^), or weight loss (when BMI is >35 kg/m^2^) are recommended. The calorie intake is suggested to be around 30 kcal/kg body weight and reduced to 25–30 kcal/kg when patients are ventilated noninvasively. In a randomized and double-blinded placebo-controlled study on the LIPCAL-ALS cohort published in 2020, a high-caloric fatty diet was examined for its effect on overall patient survival and also on vital capacity (VC), quality of life (Schedule for the Evaluation of Individual Quality of Life, SEIQoL), BMI, and appetite (Council on Nutrition Appetite Questionnaire, CNAQ) [20]. A significant increase in BMI, as well as an increased survival probability in a subgroup of fast-progressing patients (according to ALSFRS-R point-loss per month), could be objectified. It remains, however, a matter of discussion as to whether the fatty diet or the increased caloric intake is to be held responsible for this effect. Up to this day, there is no consensus on what kind of high-caloric nutritional therapy should be suggested in general.

As disease progress, bulbar symptoms, dysphagia, and general weakness for oral nutrition can worsened. In this case, initial support such as high caloric supplements, thickening of liquids and chaperoned swallowing training are not anymore able to compensate for weight loss or swallowing difficulties. Nutrition via a percutaneous gastrostomy (PEG), radiologically inserted gastrostomy (RIG), or nasogastric tube (NGT) must then be considered [19]. Given that ALS is a progressive disease and nasogastric feeding is associated with higher numbers of intervention failures, as well as a reduction of well-being resulting from, among other things, sensory irritations, NGT is primarily a bridging option until PEG or RIG are established [108,109]. Nutrition via PEG is associated with prolonged survival; the timing of PEG insertion, however, remains a matter of discussion [110,111]. Although clinical practice suggests insertion while predicted FVC is above 50%, Bond et al. have demonstrated a survival benefit for those with predicted FVC above 60% at tube placing time [110]. In addition, PEG insertion prior to significant weight loss has been associated with a prolonged tracheostomy-free survival [112]. However, there was no difference in disease duration between patients with different PEG placement timings according to their disease duration [110]. Interestingly, PEG feeding did not have a significant effect on patient mood measured by the clinical impression of mood [110].

#### 2.2.3. Ventilation

Ventilation marks one of the basic therapeutic options in the symptomatic treatment of patients affected by ALS, especially during later disease stages (Figure 1e). Most patients do not show respiratory symptoms upon diagnosis but develop diaphragm muscle wasting during the course of the disease [2]. The resulting hypercapnia can lead to sleep disorders with daytime sleepiness, fatigue, and severe impairment in quality of life [113,114].

Parameters such as FVC, VC, peak cough flow, and, in patients with bulbar symptoms, sniff nasal pressure (SNP) are easily measured upon clinical visits. Additional blood-gas analyses or nocturnal measurements of oxygen saturation can be helpful for the decision-making and monitoring of implemented NIV therapy [19]. NIV initiation is recommended, according to the EFNS guidelines, when either clinical symptoms develop or FVC drops below 80% [19]. However, positive effects can be observed in asymptomatic patients not fulfilling these criteria, and cough-assisting devices helping bronchial clearance can maximize them [19,113]. Certainly, given patient consent, the benefit of NIV therapy is undiscussed, considering prolonged survival rates in frequent users [21,115,116]. Increased survival time is strongly associated with the usage duration per day. One study found increased survival rates for patients with a minimum of four hours of NIV per day, when adjusted for confounding predictors of survival [117]. Apart from the known survival benefit, there is evidence for improved quality of life associated with NIV therapy [21,118,119].

There are remarkable regional differences concerning the rate of invasive ventilation implemented in ALS patients. While in most European countries invasive ventilation via tracheostomy is performed in about 10% of patients, studies from Asian countries often report over 20% [120]. Studies from different countries and continents consistently report prolonged survival in invasively ventilated patients and the effect is greater in younger patients, patients without signs of Frontotemporal Dementia (FTD), and spinal onset phenotypes [120,121]. Unfortunately, few studies have been performed on the quality of life after tracheostomy in ALS, which might be impaired by the need for intensive 24-h care and the initial loss of vocal communication [121]. Due to the invasive nature of this ventilation method, it is important to discuss all potential risks and benefits with patients and their caregivers prior to tracheostomy [122]. The involvement of primary caregivers in invasive ventilation has been shown to be especially important [123].

## 3. On-Going Development and Future Perspectives in the Field

### 3.1. Therapies Currently Being Tested

As various trials aiming for new drug validation are taking place at the moment, we attempt to highlight some of the latter, focusing on the different techniques and the targeted pathways [11,15,124]. In this review, ongoing trials are divided into four distinct groups: (1) development of small molecules, (2) gene-specific therapies, (3) monoclonal-antibody therapies, and (4) stem-cell therapies. The different targeted pathways and therapeutical approaches currently evaluated in clinical trials discussed in this review are resumed in Figure 2 and Figure 3 and listed in Table 1.

#### 3.1.1. Small Molecules

There are numerous ongoing studies regarding small molecule development, with a high number of them undergoing clinical phase III trials [15,124]. In this part, we will mainly focus on molecules tested in phase II/III and in phase III clinical trials to examine drugs closest to possible approval. Each of these small molecules has a direct influence on several dysregulated pathways associated with ALS. Amongst them are glutamate excitotoxicity, oxidative stress, inflammation, autophagy, and metabolism, as well as neuronal death and muscle denervation and weakness. In order to give a short overview, we tried to group molecules according to the pathway they are the most closely related to (Figure 2).

##### Excitotoxicity

As mentioned above (Section 2.1.1), a main mechanism in treatment with Riluzole is targeting excitotoxicity, especially of glutamate. Apart from that, other phase III clinical trials are also aiming to reduce excitotoxicity in ALS (Figure 2a). For example, cannabinoids have been shown to decrease spasticity, and are therefore used as symptomatic treatment in ALS [125]. Based on their protective effect on oxidative cell damage and excitotoxicity, MediCabilis CBD Oil (NCT03690791) was tested in a single-center phase III clinical trial in a small sample of participants in Australia to further evaluate a possible disease-modifying effect [126,127]. Furthermore, Memantine (NCT04302870), a noncompetitive NMDA receptor antagonist used in the management of Alzheimer’s disease, is currently being tested in a phase II/III clinical trial in the multiarm MND-SMART trial to help reduce excitotoxicity [128].

##### Oxidative Stress

Edaravone and Sodium Phenylbutyrate/Taurusodiol are, as mentioned before (Section 2.1.2 and Section 2.1.3), mainly associated with the reduction of oxidative stress.

Currently, four other drugs with this mechanism are being tested in phase III clinical trials (Figure 2b). Trazodone, an approved drug for the treatment of depression, is tested in a phase II/III trial as part of the multiarm MND-SMART trial mentioned above (NCT04302870) [128]. This drug inhibits protein kinase ribonucleic acid (RNA)-like ER kinase (PERK) and reduces the formation of stress granules [128]. In cellular and ALS animal models, the inhibition of PERK showed a reduction of TDP-43 (transactive response DNA-binding protein 43 kDa) toxicity, and Trazodone was associated with a decreased production of toxic proteins [129,130]. Furthermore, Pridopidine is a Sigma-1 receptor agonist that increases brain-derived neurotrophic factor secretion and is suggested to enhance axonal transport and promote neuronal survival in a superoxide dismutase 1 (SOD1) mouse model [131]. Pridopidine (NCT04615923) is investigated as part of the phase II/III multiregimen HEALEY ALS platform trial (NCT04297683). Deferiprone (NCT03293069), also tested in a phase II/III clinical trial, is a chelator and, therefore, is able to reduce iron accumulation and the related excessive oxidative stress in motor pathways [132]. Huolingshengji granules (NCT04950933), currently in a clinical phase II/III trial, are a traditional Chinese herbal medicine formula that consists of six herbs that demonstrate reduction of oxidative stress in ALS cell and mouse models [133,134].

##### Neuroinflammation

Among the drugs in clinical phase III targeting neuroinflammation, we can count six different trials (Figure 2c). Some trials were recently stopped in phase II/III due to their lack of benefit in ALS patients. Zilucoplan (NCT04436497), stopped in mid of June 2020, and Ravulizumab (NCT04248465), stopped in January 2023, are both inhibitors of complement component C5, a part of the immune-response pathways. One might conclude that C5 is not to be a plausible target in ALS. However, neuroinflammation as a target for new molecules includes studies on Triumeq (NCT05193994) as well. It is a repurposed drug combination of dolutegravir, abacavir, and lamivudine, used in HIV treatment and chosen for its anti-inflammatory and anti-HERV (human endogenous retroviruses) antiviral qualities, which are currently being tested in an Australian ALS multicenter phase III trial [135]. MN-166 (Ibudilast, phase IIb/III trial) (NCT04057898) is a small molecule with the ability to inhibit proinflammatory cytokines in activated glial cells and several other mechanisms targeting neuroinflammation [136]. MN-166 is further believed to enhance autophagosomal synthesis and autophagosome–lysosome fusion, as shown in a cellular study [137]. Masitinib (NCT03127267), a tyrosine kinase inhibitor targeting, in part, mast cells and microglial cells, aims to reduce neuroinflammation and modulate the neuronal microenvironment in both the central and peripheral nervous systems [138,139]. Masitinib is currently tested in a confirmatory phase III trial (NCT03127267) in a group of early disease stage patients with moderate functional impairment at baseline and is expected to have the greatest treatment effect in this group, as previously established from a phase III trial with broader inclusion criteria [138,139]. The phase II/III multiregimen HEALEY ALS Platform Trial also evaluates Verdiperstat (NCT04436510), a molecule that inhibits myeloperoxidase. In the literature, inhibition of myeloperoxidase results in reduced motor impairment and microglia activation, decreasing neuroinflammation in a mouse model [140]. According to the observation of post mortem human brain tissues, this drug could have a greater effect on patients with SOD1 mutation [141].

##### Others

Apart from excitotoxicity, oxidative stress, and neuroinflammation, other ALS-linked disease mechanisms have been targeted in clinical trials. Boosting autophagy and direct action on protein aggregation are some of the paths pursued in alternative approaches. Arimoclomol (NCT03836716), a drug that increases heat shock protein production and enhances autophagy, did not meet its primary or secondary endpoints of impact on function or survival, and the corresponding trial was terminated in July 2021. For now, not much is known about the results of this terminated study. Hopefully, the planned publication will allow insight into potential improvements and changes in future clinical trials targeting autophagy. Parallelly, other clinical trials targeting autophagy are still ongoing and look promising. Amongst those, Trehalose (NCT05136885), a disaccharide, is currently being tested as part of the HEALEY ALS Platform Trial in a phase II/III trial in the USA [142,143]. It is capable of enhancing autophagy and decreasing SOD1 mutant aggregates in mice.

One phase III trial is currently testing a fast skeletal muscle troponin activator, Reldesemtiv (NCT04944784). This drug is a small molecule designed to slow the release of calcium during muscle contraction, improving muscle function and movement. In phase IIb, a benefit for Reldesemtiv could be demonstrated over time, when measuring the revised ALS functional rating scale (ALSFRS-R) and the slow vital capacity, especially at the study endpoint of 12 weeks. The authors postulate that the magnitude of the effect of Reldesemtiv could be even greater after a longer period of treatment [144]. However, this trial has been discontinued on 31 March 2023 due to futility, as no further evidence of positive effect had been observed on the primary and secondary endpoint after 24 weeks in patients treated with Reldesemtiv when compared to placebo-treated patients.

Methylcobalamin (NCT03548311), a form of vitamin B12, when used in an ultrahigh dosage, was found to reduce denervation and reduce muscle weakness in a mouse model [39]. The results of the clinical phase III trial demonstrate similar positive results in humans with a slower functional decline in patients with early-stage ALS and having a moderate progression rate [145].

Insulin-like growth factor-1 (NCT00035815) is a neurotrophic factor essential for the nervous system’s development and can, as shown in animal models and cell culture systems, have a protective effect on neuronal cells [146,147,148,149]. This effect is, supposedly, due to the blockage of cell-death pathways and promotion of muscle reinnervation as well as axonal growth and regeneration. However, results from a two-year phase III study concluded no benefit for ALS patients concerning survival time or muscle weakness [150]. To note, this study did not investigate the effect of IGF-1 on subgroups of patients concerning, for example, familial SOD1 mutations. Indeed, to our knowledge, the positive effect of IGF-1 was not investigated in any other model than SOD1 mouse models [146,147,148,149].

CNM-Au8 (NCT04615923) is another drug investigated as part of the phase II/III multiregimen HEALEY ALS Platform Trial. This molecule is a gold nanocrystal acting on energy metabolism and was developed by Clene Nanomedicine. This drug catalyzes the oxidation of NADH into NAD+ and increases the production of ATP to reduce oxidative stress in a mouse model [151].

To conclude the part on small molecules, there are many more clinical trials currently ongoing, either in phase II or I, and not detailed in here but nicely described in other recent reviews [11,15,124]. There are high expectations from current trials of small molecules, particularly for phase II/III trials approaching several new molecules at the same time. Examples are the multiregimen HEALEY ALS Platform Trial (estimated completion date: April 2026) or the multiarm MND-SMART (estimated completion date: December 2026) trial.

#### 3.1.2. Gene-Specific Therapies

##### Antisense Oligonucleotides

Antisense oligonucleotides (ASOs) are single-stranded synthetic oligonucleotides that can be designed to complement specific mRNAs and induce their degradation. ASOs have caused widespread media attention, with the FDA approval of Nusinersen and Eteplirsen in 2016 for the treatment of spinal muscular atrophy and Duchenne muscular dystrophy that revolutionized the treatment of these two diseases [152,153]. Thus, ASOs can be a powerful treatment approach in single-gene mutation diseases and could also be beneficial in ALS patients with specific gene mutation (Figure 3a) [154]. The first phase III trial further elucidating ASOs in ALS with SOD1 mutation examined Tofersen (NCT02623699; BIIB067), a drug applied intrathecally. Based on promising subgroup analyses from phase I and II trials, the VALOR (Part C) trial was performed [155]. While the ALSFRS-R levels in ALS patients did not differ significantly between the trial and placebo groups after 28 weeks, SOD1-levels in cerebrospinal fluid (CSF) and neurofilament light chain (NfL) levels in plasma lowered significantly [156]. Adverse events were most often related to lumbar puncture [156]. Of note, in this study, participants receiving Tofersen during earlier disease stages showed a smaller decline in ALSFRS-R score than those that had started Tofersen during the open-label extension phase. Based on this observation, participants are monitored in a long-term evaluation trial with expected completion in July 2024 (NCT03070119). A study analyzing the effects of Tofersen in presymptomatic genetic SOD1 mutation carriers has been launched (ATLAS; NCT04856982) and expanded access programs enable global access for eligible patients (NCT04972487). To note, the ALS population with mutant SOD1 represents 15–20% of fALS and approximately 3% of sALS. Thus, this strategy could be beneficial to a significant part of the ALS population and represents a big step forward in ALS therapy.

Developed for patients with fused in sarcoma (*FUS*) gene mutations, ION363 is currently undergoing a phase III trial as well. The drug is applied intrathecally [157]. Autopsy results of the first application in a human suggested reduced FUS pathology due to the treatment [157]. Several other patients have therefore been treated in the expanded access program. The ongoing phase III trial (FUSION; NCT04768972), a placebo-controlled trial with a 2:1 ratio of investigational medical products to placebo, includes a “rescue” option that allows moving participants into the open-label extension phase in case of significant functional decline during the placebo-controlled study-observation period.

Further treatments that involve ASOs are currently being tested but have not reached phase III yet. A phase I clinical trial on IIB078, an ASO targeting C9orf72 (chromosome 9 open reading frame 72), has been stopped due to its lack of efficiency and reported worsening of symptoms in C9orf72 ALS patients (NCT03626012B). While SOD1 and FUS ALS patients, respectively, demonstrate an abnormally high amount of SOD1 and FUS protein, C9orf72-ALS patients, in contrast, demonstrate haploinsufficiency. In C9orf72-ALS patients, the RNA production from the C9orf72 gene, not the protein itself, is toxic [158,159]. Thus, reduction of the C9orf72 protein with ASO might not work for this subgroup of ALS patients. However, another ongoing phase I/II clinical trial has been initiated on WVE-004, using a different ASO technology. For this trial, the ASO is targeting transcriptional variants containing hexanucleotide repeat expansion without affecting the protein level of C9orf72 (NCT04931862). BIIB105 is being tested in a phase I trial for participants with Ataxin-2 (ATXN2) repeat expansion (NCT04494256). ATXN2 is usually associated with Spinocerebellar Ataxia Type 2 (SCA2), causing the disease by an expansion of CAG triplets above a length of 22–23 repeats [160]. While repeat expansions above 35 are associated with full penetrance for SCA2, Elden et al. discussed an association of intermediate lengths of 27–33 repeats with ALS [160,161]. Targeting ATXN2 induces the reduction of TDP-43 and showed improved motor function and survival in mouse models, which is why BIIB105 could be beneficial for both patients with and without Ataxin mutations, lowering the TDP-43 protein level [162].

##### Viral Vectors Delivering RNA Interference and CRISPR/Cas9

Viral vectors, for example, nonpathogenic adeno-associated viruses (AAVs), have the ability to transport exogenous molecules into cells. This mechanism can be used to deliver therapeutic agents into cells of the central nervous system (CNS). In contrast to therapeutic approaches based on ASOs, micro-RNA- or CRISPR/Cas9-containing vectors would only be injected once [163].

RNA interference is a biological mechanism initiated by the presence of short double-strand RNA (dsRNAs). Once dsRNAs are detected, they are degraded into micro-RNA (miRNA) or small interfering RNA (siRNA) and incorporated into the RNA-induced silencing complex (RISC). This RISC complex can then degrade the complementary mRNAs of the microRNA and thus prevent protein translation. Recent therapies design specific miRNAs to prevent the translation of ALS-linked proteins (Figure 3b). For example, APB-102 (also named AMT-162) a SOD1 microRNA, delivered in AAVs and applied via intrathecal injection, was tested in two human patients [164]. Once incorporated into nervous cells, APB-102 targets and binds SOD1 mRNA, subsequently reducing its protein production. Not only the misfolded mutant SOD1 but also the misfolded wtSOD1 accumulate and aggregate in both familial ALS (fALS) and sporadic ALS (sALS) [165]. Thus, by reducing global SOD1, APB-102 may slow down ALS progression by reducing aggregation, improving the survival and function of motor neurons, and potentially providing a new therapeutic opportunity for ALS patients with SOD1 pathology. This approach could potentially give results similar to treatment with ASO against SOD1. However, the benefit of this approach is its convenience, avoiding readministration by performing a single injection. UniQure is currently acquiring the rights to develop this technique for phase I/II clinical trials, which is expected for the second half of 2023. Meanwhile, the company is also developing similar techniques to treat ALS patients with C9orf72 mutations using AMT-161 miRNA molecules. However, not much is known about this future product, and regarding the negative and detrimental result obtained with ASO treatment against C9orf72, we might expect an identical outcome when targeting global C9orf72 production. In 2022, at the 29th European Society of Gene and Cell Therapy (ESGCT) Congress, UniQure also presented AAV-miQURE^®^ results in an ALS C9orf72 mouse model [166]. This strategy selectively binds the mutant repeat expansion of C9orf72 mRNA, leading to the degradation of the toxic target transcripts without affecting native C9orf72 mRNA and protein. Up to now, this technique looks like a much more promising therapy.

Furthermore, viral vectors can also carry the CRISPR/Cas9 technology [167]. This gene-editing method is associated with a design RNA guide that enables the Cas9 endonuclease to cut at a desired genomic location and remove and/or add new genetic sections. The technique is not yet used in clinical trials; however, studies aiming to target SOD1 and C9orf72 pathogenic alleles on cell and animal models have been initiated [168,169,170]. 

In summary, ASO, RNAi, and CRISPR/Cas9 represent three different approaches to targeted gene therapy, offering opportunities for drug development for ALS patients with genetic mutations. AAV vectors can be used to mediate other gene delivery, for example, genes essential for neurotrophic support, or for modulating the neuromuscular junction. Thus, gene therapy is not necessarily limited to fALS but is also investigated for sALS, as reviewed by Amado et al., 2021 [163].

#### 3.1.3. Monoclonal Antibodies

While there are no current phase III trials examining antibody treatment in ALS, there are interesting ongoing and completed phase II studies (Figure 3c). Amongst those, one study is testing the human monoclonal antibody AL001 (NCT05053035) in C9orf72-associated amyotrophic lateral sclerosis. AL001 binds the sortilin receptor and increases progranulin protein (PGRN) levels in C9orf72 ALS patients. In ALS mice models, progranulin demonstrates reduced TDP43 aggregation [171]. Furthermore, there is an ongoing phase III trial investigating AL001 in participants with heterozygous GRN (progranulin gene) mutations (INFRONT-3; NCT04374136), a causative mutation for FTD that is, in some cases, associated with ALS-FTD or pure ALS, as well [172,173].

AP-101 (NCT05039099), a human IgG1 antibody with high affinity and selective binding to misfolded SOD1 protein, is tested in a phase II trial on participants with both fALS and sALS, based on promising results from mouse models [174]. Further monoclonal antibodies are tested in phase II trials, aiming to reduce neuroinflammation (ANX005–NCT04569435, AT-1501–NCT04322149, Tocilizumab–NCT02469896). Further information is reviewed in Johnson et al., 2022, or Jiang et al., 2022 [15,124].

#### 3.1.4. Stem Cells

Stem cells, cells that—when guided into the desired direction by growth factors—can develop into any specific cell type, could function as a therapeutic approach in ALS. By developing into functioning neurons, these cells could replace damaged or apoptotic neurons. When injecting mesenchymal stem cells, the immune system could be guided towards an anti-inflammatory, neuroprotective environment by releasing different favorable soluble factors, instead of reacting to oxidative stress or degeneration by forming a proinflammatory, in parts toxic environment. There are different types of stem cells in use in preclinical and clinical trials. The most common among them are hematopoietic or mononuclear cells, neural stem cells, induced pluripotent stem cells, and mesenchymal stem cells. The latter is the most frequently investigated stem-cell type in human trials [175]. While preclinical studies showed positive effects in rodent models, mostly in presymptomatic animals, a gap for translation to an efficient trial culture in humans is yet to be filled [175]. While transient positive effects on clinical disease progression can be observed, patients demonstrated worsening in their respiratory function. Methodological challenges include (1) adequately big trial cohorts, (2) heterogeneity of eligible patient cohorts concerning their clinical and personal characteristics, (3) lack of placebo-controlled studies, and (4) the lack of publication of previously performed stem-cell trials [175]. There is a distinct need for large multicenter and placebo-controlled trials. Two studies on bone-marrow-derived mesenchymal stem cells (BM-MSC) are found in advanced trial stages in the context of ALS (Figure 3d). NurOwn^®^ is available in an expanded-access protocol for patients who were enrolled in the preceding phase III trial. While the clinical endpoint was not met in the phase III trial, subgroup analyses of patients in early disease stages (ALSFS-R ≥ 35) as well as improvements of biomarkers of neuroinflammation, neurodegeneration, and neurotrophic factor support were promising (NCT04681118) [176]. Lenzumestrocel (Neuronata-R^®^ Inj) (NCT04745299), a BM-MSC as well, is currently being tested in a phase III clinical trial with an estimated enrolment of 115 participants, based on significant therapeutic benefits lasting at least six months in patients with ALS reported in the phase II trial (NCT01363401) [177]. Beyond these, there are several other neuronal, monoclonal, and mesenchymal stem cell therapeutic strategies that are currently undergoing a phase I/II clinical trial [15,124].

In conclusion, stem cells are an established treatment method in some hematological diseases, however, with numerous treatment-related risks. Results from the two ongoing phase III clinical trials will broaden knowledge of stem cells as a treatment option for ALS. However, the general challenges in trial culture, as discussed above, will have to be addressed before considering stem cells as a viable treatment option for ALS.

**Table 1 cells-12-01523-t001:** Overview of the different therapeutic approaches discussed in this review. Sorted by addressed pathway and clinical phases (selection).

**Approved Drug**	**Pathway**	**Phase**
Riluzole	Excitotoxicity	Approved
Edaravone	Oxidative stress	Approved
Sodium phenylbutyrate/Taurursodiol	Oxidative stress	Approved
**Therapeutic being tested (selection)**	**Pathway/Approach**	**Phase**
CBD oil (NCT03690791)	Excitotoxicity	Phase III
Memantine (NCT04302870)	Excitotoxicity	Phase II/III
Trazodone (NCT04302870)	Oxidative stress	Phase III
Huolingshengji granules (NCT04950933)	Oxidative stress	Phase III
Pridopidine (NCT04615923)	Oxidative stress	Phase II/III
Deferiprone (NCT03293069)	Oxidative stress	Phase II/III
Triumeq (NCT05193994)	Neuroinflammation	Phase III
Masitinib (NCT03127267)	Neuroinflammation	Phase III
MN-166/Ibudilast (NCT04057898)	Neuroinflammation	Phase II/III
Verdiperstat (NCT04436510)	Neuroinflammation	Phase II/III
Ravulizumab (NCT04248465)	Neuroinflammation	Stopped in January 2023
Zilucoplan (NCT04436497)	Neuroinflammation	Stopped in June 2020
Methylcobalamin (NCT03548311)	OTHERS Reduces denervation and muscle weakness	Phase III
Insulin-like growth factor (NCT00035815)	OTHERS Blockage of cell death pathways	Phase III
Trehalose (NCT05136885)	OTHERS Enhances autophagy, decreases SOD1 aggregates	Phase II/III
CNM-Au8 (NCT04615923)	OTHERS Energy Metabolism	Phase II/III
Reldesemtiv (NCT04944784)	OTHERS Slows calcium release	Stopped in March 2023
Arimoclomol (NCT03836716)	OTHERS (Increased heat shock protein expression, enhancement of autophagy)	Stopped
Tofersen/BIIB067 (NCT04972487)	Gene specific–ASO, SOD1-mutations	Phase III
Jacifusen/ION363 (NCT04768972)	Gene specific–ASO, FUS-mutations	Phase III
WVE-004 (NCT04931862)	Gene specific–ASO, mutant C9orf72	Phase I/II
BIIB105 (NCT04494256)	Gene specific–ASO, ATXN2	Phase I
IIB078 (NCT03626012B)	Gene specific–ASO, C9orf72	Stopped
APB-102	SOD1 microRNA	Phase I/II (begin date mid–end 2023)
AMT-161	C9orf72 microRNA	Substance in development
AAV-miQURE	Mutant C9orf72 microRNA	Substance in development
AL001 (NCT05053035)	Monoclonal AB, C9orf72	Phase II
AP-101 (NCT05039099)	Monoclonal AB, SOD1	Phase II
ANX-005 (NCT04569435)	Monoclonal AB	Phase II
AT-1501 (NCT04322149)	Monoclonal AB	Phase II
Tocilizumab (NCT02469896)	Monoclonal AB	Phase II
NurOwn (NCT04681118)	Stem cells	Phase III
Lenzumestrocel (NCT04745299)	Stem cells	Phase III
Blood-Brain Barrier Opening Using MR-Guided Focused Ultrasound (NCT03321487)	Blood-Brain Barrier Opening	Not Applicable
Fasudil (NCT03792490)	Post-translational modifications	Phase IIa
BIIB100 (NCT03945279)	Nucleocytoplasmic transport	Stopped

### 3.2. Novel Therapeutic Approaches (Unaddressed or Poorly Addressed Yet)

We previously discussed the different clinical trials including molecules or techniques targeting certain ALS pathways. However, some ALS-dysregulated pathways are still unaddressed. In this section we will highlight different unaddressed mechanisms, amongst them post-translational modification (PTM), axonal transport, and others, discussing their potential benefit for future ALS treatment. These pathways are represented in green in Figure 4.

#### 3.2.1. Post-Translational Modification

Post-translational modifications (PTMs) impact proteins involved in the neuropathology of ALS, including FUS, SOD1, and TDP-43 [178,179,180]. Changes in acetylation, methylation, or phosphorylation are known to influence not only protein function but also their subcellular locations, interaction with other proteins or RNA, liquid-liquid phase separation, stress granule formation, cell death, as well as other processes [180,181,182,183,184]. A few strategies to target post-translational modification can be cited (Figure 4a).

Epigenetic mechanisms, including DNA methylation and histone modifications, are part of these PTMs and function as a key process in gene transcription regulation and DNA damage repair [185,186,187,188,189]. Recent studies have reported several dysregulations in histone PTMs in numerous ALS models, including alterations in acetylation, methylation, and phosphorylation, as well as global changes in epigenetic enzyme modifiers [190,191,192]. In order to revert these disease-associated alterations, most studies focus on the use of histone deacetylase inhibitors (HDACi) [193]. For example, the HDAC6 inhibitor is able to restore axonal transport and metabolic-lipidomic functions in ALS-FUS and ALS-TDP43 iPSC (induced pluripotent stem cells)-derived motor neurons [194,195,196,197]. However, delivery to target and drug tolerance of HDACi are still improvable, as some show cytotoxic effects, depending on the cell type [198]. Interestingly, sodium phenylbutyrate, an HDACi, was recently approved by the FDA in combination with Taurursodiol, although the underlying mechanism in this context remains unclear [3]. Other studies report promising results when targeting protein and histone phosphorylation with commercially available kinase inhibitors, improving TDP43 localization and aggregation [182,192,199,200]. Fasudil, a kinase inhibitor, is currently being tested in a phase II clinical trial for ALS treatment (NCT03792490). As observed in ALS mouse models, it was able to restore the phosphorylation levels of certain proteins associated with synaptic and neuroinflammatory functions [201,202].

Hope et al. recently detected changes in DNA methylation patterns in blood samples of more than 6.000 ALS patients worldwide and observed, that Riluzole was not able to revert specific DNA methylation changes in an efficient manner [203]. Motor neurons in human ALS show a significant increase in so-called DNA methyltransferases (DNMTs), responsible for DNA methylation production, as well [204]. Thus, inhibition of aberrant DNA methylation is a new direction that could possibly become relevant for new therapies for ALS. Some of these DNMT inhibitors are resumed by Martin et al., 2013 [205].

#### 3.2.2. Axonal Transport

Targeting axonal transport is not completely unaddressed in clinical trials. Pridopidine (NCT04297683), demonstrating enhanced axonal transport in a SOD1 mouse model, is currently being tested in a phase II/III trial [131]. As previously described, HDAC6 inhibitors can restore axonal transport defects in patient-derived motor neurons with FUS or TDP43 mutations. They are also able to restore axonal ER transport in the iPSC FUS model and mitochondrial transport in both FUS and TDP43 iPSC models [194,197]. Interestingly, these two studies also demonstrated that HDAC6 inhibitors restore several other pathways altered in ALS, amongst them metabolic-lipidomic functions, aggregation, and subcellular localization. Since axonal transport interacts with other pathogenic mechanisms, restoring the latter might have an overall positive effect on disease progression (Figure 4b) [206].

#### 3.2.3. Nucleocytoplasmic Transport

Nucleocytoplasmic transport is poorly addressed in clinical trials but widely altered in ALS [207]. Inhibition of nuclear export or enhancement of nuclear import could be strategies to restore nucleocytoplasmic function in ALS (Figure 4c) [208,209,210,211]. The only clinical trial examining nucleocytoplasmic transport, that we are aware of, was implemented by Biogen. Here, BIIB100, a nuclear-export inhibitor targeting XPO1 (NCT03945279), was tested; however, its phase I trial was terminated in June 2021 without clear reason. 

#### 3.2.4. DNA Damage

While the evidence linking excessive DNA damage to ALS is increasing, clinical trials acting on this pathway are lacking [212]. Modulating, for instance, the Poly ADP-ribose polymerase PARP1, a protein predominantly triggered by DNA damage, could have a positive effect on the disease. Generally, PARPs are enzymes that catabolize NAD+ to sequentially add adenosine diphosphate (ADP)-ribose subunits onto target proteins. This generates polymers of poly (ADP-ribose) (PAR). Interestingly, increased PAR was observed in the motor neurons of an ALS spinal cord [213]. Inhibition of PARP via Veliparib, an FDA-approved drug, decreases TDP-43 aggregation in culture cells and reduces TDP-43-associated neuronal loss in rat spinal cord cultures [213]. Olaparib, another FDA-approved PARP inhibitor, demonstrated a reduction of PAR levels and rescued TDP-43-induced death in culture cells. There are other approved PARP inhibitors that could be tested in the context of ALS [214]. Apart from PARP inhibitors, Fasudil and Cannabidiol (CBD), both previously discussed, are able to reduce the expression of PARPs (Figure 4d) [215,216].

#### 3.2.5. Molecular Tweezer

Currently, most of the strategies regarding the inhibition of protein aggregation rely on the use of monoclonal antibodies, as previously discussed with AL001 (NCT05053035) and AP-101 (NCT05039099) clinical trials. A new interesting approach might be the use of small molecules, called molecular tweezers, which help in preventing the aggregation of toxic proteins (Figure 4e). Molecular tweezers are host molecules with open cavities capable of binding molecules. For example, the CLR01 molecular tweezer acts as a nano chaperone that transiently binds abnormal protein self-assembly at positively charged amino acid residues (primarily lysine and to a lower extent arginine) and inhibits aggregates of multiple disease-associated proteins [217]. Thus, molecular tweezers work as a tool to target the process of abnormal protein self-assembly itself rather than a particular protein. The molecular tweezer CLR01 is able to decrease SOD1 aggregation in vitro and in vivo in the SOD1 mouse model [218,219]. Unfortunately, this strategy did not succeed in slowing down motor symptoms. In contrast, CLR01 inhibiting Tau aggregation was able to ameliorate muscle-strength deterioration and anxiety- and disinhibition-like behavior in an Alzheimer’s disease mouse model [220]. To our knowledge, CLR01 has not been tested in other ALS models than SOD1. CLR01 action in SOD1 is mainly explained by binding to lysine residues, and results vary depending on the type of mutation [218]. Since CLR01 also acts through arginine binding, and that FUS contains different arginine-rich regions, it is then tempting to speculate that CLR01’s impact on the FUS-ALS model might have a different outcome than the one observed in the SOD1-ALS model. Neither was CLR01 tested in the C9orf72 model to test its efficiency on PR (proline-arginine), or GR (glycine-arginine) dipeptide repeat aggregation.

## 4. Stratification towards Personalized Medicine

When discussing clinical trials, it is important to discuss their current practices and potential modification.

First, most preclinical trials on mouse models have been tested in SOD1 models before transitioning to human testing [12,13]. Due to numerous failed studies in human clinical trials based on SOD1 mouse models, this practice is facing much criticism considering that a majority of human ALS cases are sporadic, with unknown genetic causes; a single animal model with a focus on one mutation may not be sufficient. A better understanding of animal models and the determination of specific, transitional models applicable to the human might be the key to improving clinical-trial culture.

Secondly, recent clinical trials enroll lower numbers of participants, which might impact statistical significance [14]. In order to ensure comparability, a standardization of primary outcomes and standards in treatment efficacy monitoring are needed. As a heterogenous disease, ALS is caused by a conjunction of different factors. Environmental, genetic, and age-related alterations ultimately lead to cell death in upper and lower motor neurons and consequently impairment in different body regions [2]. The clinical presentation and course of the disease, the possible genetic causes, as well as the underlying neuropathology and the treatment responses differ greatly. Thus, it is legitimate to wonder if “the one” ALS might not exist. This heterogeneity could, at least partly, explain the reduced statistical power in clinical trials, and indicate a need for a more personalized therapy in ALS [221]. Most larger pharmacological trials address a broad, heterogeneous cohort. Even when adjusting for known confounders, a matter of discussion remains whether trials should be performed differently in the future as “one size does not fit all” [222].

### 4.1. Clinical Characteristics

As a first step towards personalized trial culture, future studies will benefit from analyzing results depending on ALS subgroups. Overlaps of ALS with other diseases such as FTD or parkinsonism offer further possibilities for stratification [2]; 30–50% of ALS patients develop behavioral impairment, or criteria for FTD which is associated with worsened disease progression and shorter lifespan [1,223,224]. The phenotype, as well as overlap syndromes, show remarkable differences in overall survival. The pure lower motor neuron phenotype, for instance, has a median survival more than three times longer than the bulbar phenotype [225]. Multiple studies have identified the clinical phenotype as an independent prognostic factor and some have, at least in parts, included the phenotype as a prognostic factor into their model for prognosis [226,227,228]. Previous clinical trials demonstrated differences in treatment effectiveness depending on clinical phenotypes defined by disease onset or progression rate [4,20]. Future studies should consider stratifying groups by their phenotypic manifestation. This might enhance the prediction of clinical progression rate (fast vs. slow), possibly resulting in improved statistical power.

### 4.2. Genetic Classification

Another strategy for subgroup division is stratification according to genetic mutations (for example, see Section 3.1.2 on gene-specific therapies). Overall, there are more than 30 genes known to cause fALS. The most common mutations affect C9orf72, SOD1, FUS, and TDP-43, differing in both clinical manifestation as well as treatment response [229]. Currently, so-called sporadic cases are occasionally not admitted to genetic testing, even though an empty family history does not exclude genetic alterations [229]. Not only de novo mutations, but also incomplete penetrance and phenotypic variability, presymptomatic disease stages in relatives, or even earlier, non-ALS-related death of mutation carriers can cause a falsely negative family history [230]. Grassano et al. found genetic alterations in 21.5% of sALS cases in a cohort of 1043 ALS patients and a genetic diagnosis in almost 27% of all individuals after having performed whole-genome sequencing [231]. It seems desirable to offer genetic counseling and routine genetic testing to all ALS patients at the time of diagnosis. Importantly, the results of genetic testing may require long waiting times, delaying possible treatment. Acceleration of genetic testing should be focused upon. When genetic alterations are present, relatives at risk may be offered genetic counseling as well as predictive testing. Presymptomatic monitoring and treatment possibilities for asymptomatic carriers may offer hope for thus far clinically unaffected individuals (e.g., ATLAS study for presymptomatic SOD1-mutation carriers—NCT04856982) [232]. As it is often suggested that disease evolution is gradual, presymptomatic treatment may prolong time-to-symptom onset as well as overall survival [233]. Importantly, all findings gained through genetic testing require interpretation by experienced professionals, calling for close interdisciplinary cooperation between geneticists and neurologists as well as extensive genetic counseling prior to testing.

### 4.3. The Power of Biomarkers

While studies on biomarker discovery mainly aim to accelerate and improve ALS diagnostics, enhancements in this field may also help in the prognostic evaluation as well as treatment-response monitoring. ALS patients experience progressive functional decline and limitations. The window for therapeutic opportunities remains relatively small. Therefore, early diagnosis is the key to early access to treatment and interventions. Here, a disease-specific biomarker (or biomarker panel) with high sensitivity and specificity for ALS would revolutionize diagnosis. The use of biomarkers for diagnostic and prognostic purposes in ALS is widely discussed in the review, which resulted from the Second ALS Young Investigators Academy [234]. In this section, we mainly address biomarkers that could be useful for treatment monitoring and patient stratification. Thus far, neurofilaments (Nf), proteins that can be found in myelinated axons, have been a matter of discussion as a main biomarker in ALS. The most frequently used blood-based biomarker is NfL [235]. In the case of neuronal/axonal damage, NfL levels increase in both CSF and blood and are highly correlated [236,237]. There are efforts to establish NfL for monitoring pharmacodynamic treatment effects as well as prognostic classification. High levels of NfL have negative prognostic connotations [238,239]; however, NfL levels remain relatively stable after disease onset, complicating its use as a prognostic marker for individual disease course after onset [238,240,241,242]. In the recent phase III Tofersen trial, the primary clinical endpoint represented by ALSFRS-R decline was not reached, while significantly reduced NfL levels were observed. Nevertheless, a reduction of NfL level could be a sign of deceleration of disease progression [156]. For prognostic purposes as well as treatment monitoring, a combination of NfL with miRNA-181 derived from blood samples showed promising results [243]. Another example is TDP-43, taken from peripheral blood, which was correlated with time of generalization from bulbar or spinal symptoms to both regions. CSF pTDP-43 levels correlate with ALSFRS-R [244]. Furthermore, regulatory T cells have been investigated as markers of immune dysregulation with an inverse correlation to disease progression in ALS [245,246,247]. It may be interesting to investigate the regulatory T cells reaction to specific ALS treatments. Neopterin, a marker of proinflammatory state immune response, as well as p75, a marker of microglia and macrophage activation, both taken from urine samples, could be correlated to disease progression [248,249]. Both biomarkers are especially interesting, as urine sampling is noninvasive and therefore feasible even in advanced disease stages. Similarly, tear fluid can be sampled noninvasively and has been the subject of a recent trial with promising results [250]. However, more studies in larger cohorts and longitudinal subsets are necessary in order to evaluate the benefit for therapy monitoring through biomarkers.

## 5. Conclusions

With the number of ALS cases expected to rise to nearly 380,000 worldwide by 2040, in part due to aging populations, the optimization of current treatment and possible new treatment options appear to be of acute relevance [251]. Efforts regarding improved drug administration and delivery, as well as new target strategies, should be further addressed. The combination of one or several of the stratification methods (genetic, molecular, and clinical) will personalize the treatment of individuals and, potentially, optimize treatment response. Considering genetic testing for common genetic causes of ALS appears to be one of the first steps towards this, let alone for gene-specific treatment options such as Tofersen. An early treatment start will presumably be crucial for most future therapies, resulting in the urge for early patient identification. Apart from the development of highly sensitive and specific biomarkers, pre-/asymptomatic carriers might be identified by offering genetic testing to other family members at risk of familial/genetic ALS. Until a satisfactory causative treatment is found, supportive therapy remains of the utmost importance during disease accompaniment. In this field, more studies are necessary in order to objectify the relevant effects. The vast number of substances currently tested, as well as the rising number of clinical trials within the last years, give high hopes for future therapeutic options to treat, or possibly cure, ALS.

## Figures and Tables

**Figure 1 cells-12-01523-f001:**
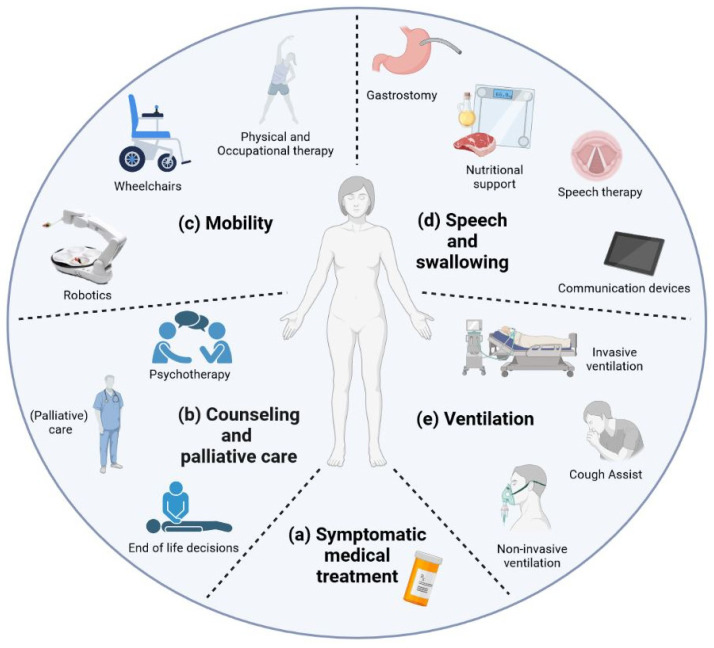
Schematic overview of supportive therapy for patients with ALS: Different aspects are highlighted: (**a**) Symptomatic medical treatment, (**b**) Counseling and palliative care, (**c**) Mobility, (**d**) Speech and swallowing, and (**e**) Ventilation. Created by Meret Herdick with “BioRender.com”.

**Figure 2 cells-12-01523-f002:**
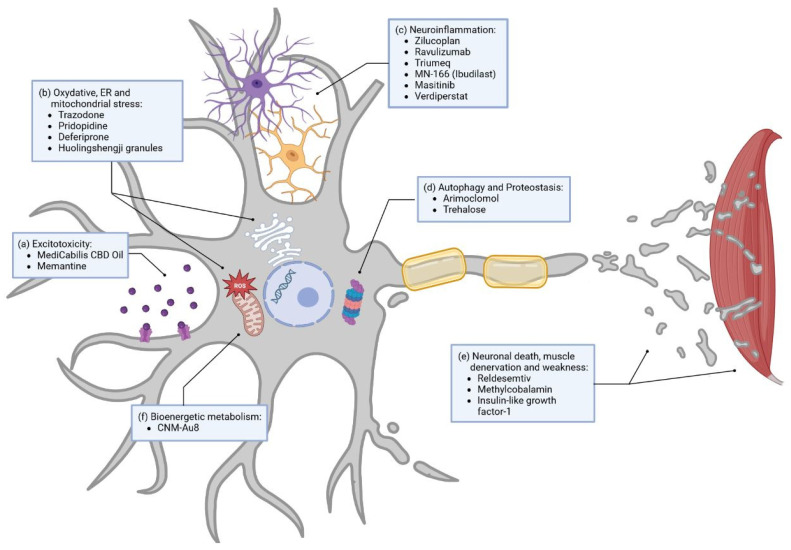
Small molecules in phase II/III and III clinical trials, grouped by mode of action: (**a**) Excitotoxicity, (**b**) Oxidative, ER, and mitochondrial stress, (**c**) Neuroinflammation, (**d**) Autophagy and Proteostasis, (**e**) Neuronal death, muscle denervation and weakness, and (**f**) Bioenergetic metabolism. ER: Endoplasmic reticulum. Created by Laura Tzeplaeff with “BioRender.com”.

**Figure 3 cells-12-01523-f003:**
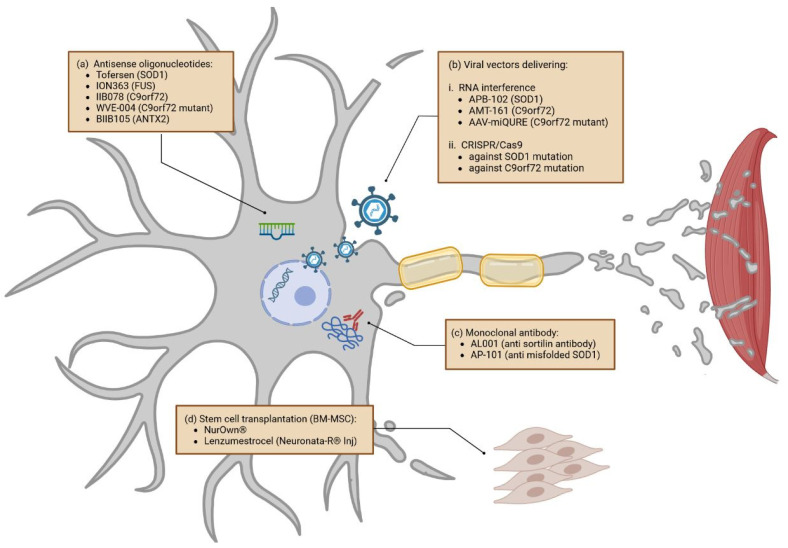
Gene, antibody, and stem-cell therapeutic approaches: (**a**) Antisense oligonucleotides, (**b**) Viral vectors delivering RNA interference or CRISPER/Cas9, (**c**) Monoclonal antibody, and (**d**) Stem cell transplantation. ANTX2: Ataxin-2; BM-MSC: bone-marrow-derived mesenchymal stem cells; C9orf72: Chromosome 9 open reading frame 72; FUS: Fused in sarcoma; SOD1: Superoxide dismutase 1. Created by Laura Tzeplaeff with “BioRender.com”.

**Figure 4 cells-12-01523-f004:**
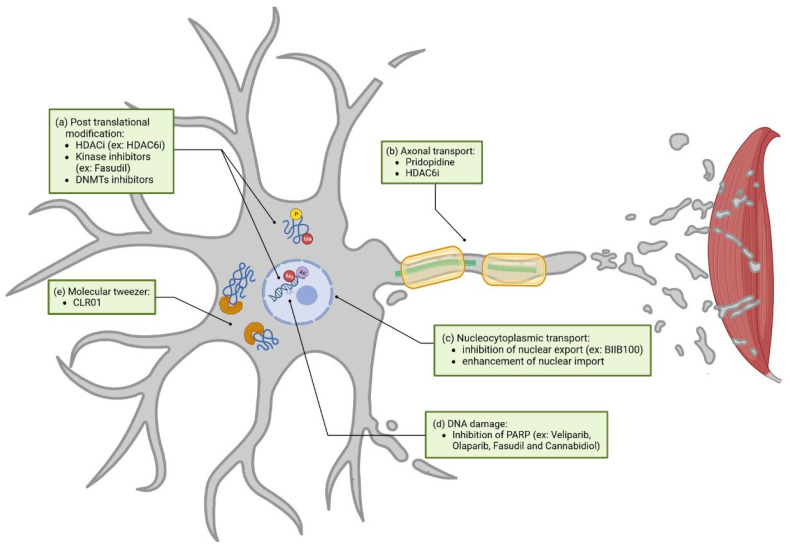
Future strategies and approaches: (**a**) Post-translational modification, (**b**) Axonal transport, (**c**) Nucleocytoplasmic transport, (**d**) DNA damage, and (**e**) Molecular tweezer. DNMTs: DNA methyltransferases; HDACi: histone deacetylase inhibitors; PARP: Poly ADP-ribose polymerase. Created by Laura Tzeplaeff with “BioRender.com”.

## Data Availability

No new experimental data were created or analyzed in this study. Data sharing is not applicable to this article.

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
