# Peer review of "Current State and Future Directions in the Therapy of ALS"

_cells, 2023, doi:10.3390/cells12111523_

Round 1

Reviewer 1 Report

In this manuscript the authors present extensive recent information about therapeutic strategies for ALS. Whereas the provided information is very relevant, some topics are not clearly within the scope of a cellular and molecular journal, e.g., "2.2. Supportive therapy". Furthermore, it would be important to organize the information provided throughout the manuscript and clearly specify the scope of the manuscript as well as the topics to be addressed in the last paragraph of the "Introduction" section. Another aspect is that probably it would be clearer for the reader that a section on disease mechanisms appears earlier in the manuscript to provide the basis for all the therapeutic strategies presented.

Author Response

  • In this manuscript the authors present extensive recent information about therapeutic strategies for ALS. Whereas the provided information is very relevant, some topics are not clearly within the scope of a cellular and molecular journal, e.g., "2.2. Supportive therapy".

Thank you very much for this comment. Given the fact that “Cells” is a journal focused on cellular and molecular topics, this is a justified critique.

Our review highlights the current state and future directions in the therapy of ALS. In this review we focus on cellular and molecular topics regarding therapeutic strategies with an extensive description of approved medication, ongoing developments (regarding small molecules, gene specific therapies, …).

Nevertheless, in the clinical setting, ALS therapy is mainly focused on symptomatic treatment. Disease modification with medical treatment only modestly slows down disease progression and increases survival time by about 4 months. Supportive therapy and care are an important aspect of disease management. We believe ALS medical treatment and supportive therapy to, currently, be inseparable. Therefore, a review on “ALS therapy” would, in our opinion, be incomplete without mentioning the complete spectrum of clinical ALS treatment.  

  • Furthermore, it would be important to organize the information provided throughout the manuscript and clearly specify the scope of the manuscript as well as the topics to be addressed in the last paragraph of the "Introduction" section.

To clearly specify the scope of the manuscript as well as the choice of including supportive therapy, we reworded the introduction section that now includes the following passage beginning in (current) line 87:

“In this review, we summarize the current state of ALS therapy. We extensively discuss medical as well as supportive therapy options as they are both important aspects of ALS disease management and, in our opinion, inseparable in the current state of ALS treatment. We further discuss the ongoing clinical trials with a particular focus on small molecules, but also clinical trials focusing on gene specific therapy (antisense oligonucleotides and viral vectors delivering RNA interference and CRISPR/Cas9) as well as monoclonal antibody and stem cell therapy. Different ALS mechanisms that are currently not targeted in clinical trials are addressed. Lastly, we highlight the current difficulties of validating new drugs in clinical trials. We suggest stratification of ALS patients considering clinical characterizations, genetics, and current studies on ALS-linked biomarkers to possibly improve ALS therapy towards personalized medicine.”

  • Another aspect is that probably it would be clearer for the reader that a section on disease mechanisms appears earlier in the manuscript to provide the basis for all the therapeutic strategies presented.

In order to answer the reviewer’s comment regarding a new introduction section on disease mechanisms, we expanded the introduction that now includes the following paragraph beginning in (current) line 43:

“Oxidative stress, excitotoxicity, mitochondrial and proteasomal dysfunctions, RNA metabolism, altered synaptic function, disturbed axonal transport and neuroinflammation are, among other pathways, considered as important contributors to ALS disease development [10,11]. However, oxidative stress and excitotoxicity are the only two pathways targeted by the FDA approved drugs. Therefore, targeting other disease mechanisms may be key to the identification of new therapeutic drugs for additive or more effective treatment of ALS. “

Reviewer 2 Report

This is a very comprehensive review on different aspects of therapy of ALS. Authors elaborated many aspects of therapy (some would rather be called "care") out of which only one is medical treatment. It is the opinion of this reviewer that it is the medical treatment that deserves an absolute focus in this article not just because of the scope of the journal (cell biology, molecular biology, and biophysics) but since the field of ALS research needs a sharp tool to tackle this still uncurable disease. In this respect more emphasis should be given to future therapies and ongoing clinical research than the current (not favorable) state with approved drugs.

Personalized medicine and patient stratification are justly emphasized, however more on the clinical but less on the biomarker side,

Generally, the paper lacks a more critical review of existing therapies and developments.

Particular comments

- The approved and applied drugs Riluzole and Edaravone (with very modest effectivness) are mentioned twice - as current treatment and under 3. Ongoing development ... (but again there in "3.1.Therapies currently being tested). The authors should decide to discuss current therapies under one chapter.

- line 436 on "rarely or even unaddressed pathways" needs to be elaborated with more critical comments. Same for the next paragraph line 440 (failure to demonstrate effectivness of a drug enhancing autophagy), or line 540 - how do authors explain the effectivness of APB-102 that target SOD1 and not its particular mutated form that induces ALS?

- Monoclonal antibodies and Stem cells therapies are placed inadequately under Gene specific therapies. The final paragraph of this chapter (#3.1.2.) starts by concluding on small molecules which are tackled in a previous chapter (#3.1.1.). Here again (lines 608-611) Riluzole and Edaravone are emphasized but also mentioned are other unnamed "similar" drugs in Phase III that we believe should deserve some more emphasis.

- Chapters 3.2 and 3.3 are not in the scope of the journal. 

- Title of 3.4.1. should state Post-translational modification instead of protein translational modification (also check other places in the ms)

- lines 695-697: what is the purpose of this sentence? Do the authors imply that some drugs were meant to tackle methylation but unsuccessfully, and if so can they suggest some drug design?

- Why is chapter 3.4.3. in bold italic 

- Chapter 3.4.4. is missing.

- Line 732-: some explanation should be given to the mechanism of action of molecular tweezers.

- Lines 741-742 should be critically elaborated.

-   Line 793: UBQLN2 and OPTN are not among most common mutations in fALS.

- It is not positive to site reviews in a review (e.g. lines 605, 642, 784-785, 821-822)

-line 856-857: Comprehensive genetic testing may not be a feasible and economical solution. Emphasis should be on finding early biomarkers. 

Author Response

This is a very comprehensive review on different aspects of therapy of ALS. Authors elaborated many aspects of therapy (some would rather be called "care") out of which only one is medical treatment. It is the opinion of this reviewer that it is the medical treatment that deserves an absolute focus in this article not just because of the scope of the journal (cell biology, molecular biology, and biophysics) but since the field of ALS research needs a sharp tool to tackle this still uncurable disease. In this respect more emphasis should be given to future therapies and ongoing clinical research than the current (not favorable) state with approved drugs.

Personalized medicine and patient stratification are justly emphasized, however more on the clinical but less on the biomarker side,

Generally, the paper lacks a more critical review of existing therapies and developments.

Dear Reviewer, we agree that we did not focus on biomarkers in our review. One of the reasons is, in fact, that this review was created as a result of a collaborative work that took place during the second ALS Young Investigators Academy in October 2022. We aimed to publish it back-to-back with the citation [Previous 228 and actual 234]. While our group and our review extensively focus on therapy, the other group’s work focused on diagnosis and discussed, in this context and in grand detail, the current ALS biomarkers. Therefore, we  decided to solely discuss biomarkers that, to our knowledge, have the potential to help in prognostic evaluation as well as treatment response monitoring. We hope you understand our decision to not discuss biomarkers in detail and kindly ask interested readers to refer to this review for more information on ALS and its biomarkers as well as clinical classification.

Particular comments

- The approved and applied drugs Riluzole and Edaravone (with very modest effectivness) are mentioned twice - as current treatment and under 3. Ongoing development ... (but again there in "3.1.Therapies currently being tested). The authors should decide to discuss current therapies under one chapter.

In agreement with the reviewer, we regrouped all information related to Riluzole, Edaravone as well as Sodium Phenylbutyrate and Taurursodiol in part “2.1. Medications”. Here we restructured all information on these three drugs in order to give a more comprehensive overview.

- line 436 on "rarely or even unaddressed pathways" needs to be elaborated with more critical comments. Same for the next paragraph line 440 (failure to demonstrate effectiveness of a drug enhancing autophagy), or line 540 - how do authors explain the effectiveness of APB-102 that targets SOD1 and not its particular mutated form that induces ALS?

For line 436:

Thank you, the sentence has been changed. We were not intending this amount of criticism.

“Apart from excitotoxicity, oxidative stress and neuroinflammation other ALS-linked disease mechanisms have been targeted in clinical trials."

line 440:

We agree with the reviewer that more information on this clinical trial and comparison with other ongoing trials targeting autophagy would be of great interest, possibly enabling discussions on reasons for Arimoclomol treatment failure. However, information on this terminated clinical trial are lacking as results have, to our knowledge, not been published. The developing company, Orphazyme, claimed that they will present the final data on the trial at the European Network to Cure ALS (ENCALS) annual meeting, July 12-14, 2023 and plan to publish this results later this year.

To answer the comment, the next paragraph has been reworked and completed with additional information:

“Boosting autophagy and direct action on protein aggregation are some of the paths pursued in alternative approaches. Arimoclomol (NCT03836716), a drug that increases heat shock protein production and enhances autophagy, did not meet its primary or secondary end points of impact on function or survival and the corresponding trial was terminated in July 2021. For now, not much is known about the results of this terminated study. Hopefully, the planned publication will allow insight into potential improvements and changes in future clinical trials targeting autophagy. Parallelly, other clinical trials targeting autophagy are still ongoing and look promising. Amongst those, Trehalose (NCT05136885), a disaccharide, is currently being tested as part of the HEALEY ALS Platform Trial in phase II/III trial in the USA [137,138]. It is capable of enhancing autophagy and decreasing SOD1 mutant aggregates in mice.”

For line 540:

Mutant SOD1 accumulates and aggregates in familial ALS. However, misfolded forms of human wtSOD1 have also been detected in post-mortem CNS tissue of both familial and  sporadic ALS without SOD1 mutations (Paré et al., 2018, PMID: 30242181) [165]. Furthermore, studies suggest that  human wtSOD1 and mutant SOD1 colocalize and aggregate together and that the human wtSOD1 allele has potential to influence the toxicity of mutant human SOD1 and the formation of aggregates (Prudenccio et al., 2010, PMID: 20871097). APB-102 is a recombinant AAVrh10 vector that expresses an anti-SOD1 artificial microRNA. The microRNA binds to SOD1 mRNA, reducing the production of global SOD1 protein. This therapy may slow down or reverse ALS progression in patients with SOD1 pathology.

In order to answer the reviewer’s comment regarding more critical comments, we reworded the paragraph on APB-102:

“For example, APB-102 (also named AMT-162) a SOD1 microRNA, delivered in AAVs and applied via intrathecal injection was tested in two human patients [164]. Once incorporated in nervous cells, APB-102 targets and binds SOD1 mRNA, subsequently reducing its protein production. Not only the misfolded mutant SOD1 but also the misfolded wtSOD1 accumulate and aggregate in both fALS and sALS [165]. Thus, by reducing global SOD1, APB-102 may slow down ALS progression by reducing aggregation, improving survival and function of motor neurons and potentially provide a new therapeutic opportunity for ALS patients with SOD1 pathology. This approach could potentially give results similar to treatment with ASO against SOD1. However, the benefit of this approach is its convenience, avoiding readministration by performing a single injection. UniQure is currently acquiring the rights to develop this technique for Phase I/II clinical trials, which is expected for the second half of 2023.”

- Monoclonal antibodies and Stem cells therapies are placed inadequately under Gene specific therapies. The final paragraph of this chapter (#3.1.2.) starts by concluding on small molecules which are tackled in a previous chapter (#3.1.1.). Here again (lines 608-611) Riluzole and Edaravone are emphasized but also mentioned are other unnamed "similar" drugs in Phase III that we believe should deserve some more emphasis.

In accordance with this remark, we renumbered Monoclonal antibodies and Stem cells therapies titles into  “3.1.3. Monoclonal Antibodies” and “3.1.4. Stem cells”.

Also, we removed what aimed to be a general conclusion, but was already discussed before or later in the discussion part and only kept what was related to stem cells and reworked it:

"In conclusion stem cells are an established treatment method in some hematological diseases but with many treatment-related risks. Results from the two ongoing phase III clinical trials will broaden knowledge on stem cells as a treatment option for ALS. However, the general challenges discussed above will have to be addressed before considering stem cells as a viable treatment option for ALS."

- Chapters 3.2 and 3.3 are not in the scope of the journal.

Thank you for your comment. We agree that the former part 3.3., “New strategies to enhance drug delivery”, may be beyond the scope of this journal. We have therefore decided to remove this part.

To comprehensively introduce Riluzole, Edavarone and PB/TURSO in chapter 2.1., we have included some parts of former chapter 3.2. “Improvements of drug administration”. The development of liquid solutions, for example for patients with dysphagia, as well as oral treatment options as opposed to intravenous administration are of great clinical importance and need to be mentioned when introducing the three aforementioned drugs.

- Title of 3.4.1. should state Post-translational modification instead of protein translational modification (also check other places in the ms)

Thank you for this comment, we have changed the manuscript, accordingly.

- lines 695-697: what is the purpose of this sentence? Do the authors imply that some drugs were meant to tackle methylation but unsuccessfully, and if so can they suggest some drug design?

We agreed that this sentence might be a bit confusing. we suggest this new formulation:

“Hope et al recently detected changes in DNA methylation patterns in blood samples of more than 6.000 ALS patients worldwide and observed, in fact, that Riluzole was not able to revert specific DNA methylation changes in an efficient manner [203]. Motor neurons in human ALS also show significant increase of so-called DNA methyltransferases (DNMTs), responsible for DNA methylation production [204]. Thus, inhibition of aberrant DNA methylation is a new direction that could be relevant for new therapies of ALS. Some of the DNMT inhibitors that would possibly be relevant for ALS are resumed in Martin et al., 2013 [205].”

- Why is chapter 3.4.3. in bold italic + - Chapter 3.4.4. is missing.

Thank you for these helpful comments, the mistakes were corrected.

- Line 732-: some explanation should be given to the mechanism of action of molecular tweezers.

In accordance with the reviewer comment, a few sentences to explain the mechanism of action of molecular tweezer was added 3.4.6:

“3.4.5. Molecular tweezer

Currently, most of the strategies regarding inhibition of protein aggregation rely on the use of monoclonal antibodies, as previously discussed with AL001 (NCT05053035) and AP-101 (NCT05039099) clinical trials. A new interesting focus might be the use of small molecules, called molecular tweezers, which help prevent the aggregation of toxic proteins. Molecular tweezers are host molecules with open cavities capable of binding molecules. For example, the CLR01 molecular tweezer acts as a nano chaperone that transiently binds abnormal protein self-assembly at positively charged amino acid residues (primarily Lysine and to a lower extent Arginine) and inhibits aggregates of multiple disease-associated proteins [217]. Thus, molecular tweezers work as a tool to target the process of abnormal protein self-assembly itself rather than a particular protein. In the context of ALS, CLR01 is able to decrease SOD1 aggregation in vitro and in vivo in the SOD1 mouse model [218,219]. However, this strategy did not succeed in slowing down motor symptoms. In contrast, CLR01 inhibiting Tau aggregation was able to ameliorate muscle-strength deterioration, anxiety-, and disinhibition-like behavior in a Alzheimer disease mouse model [220]. “

- Lines 741-742 should be critically elaborated.

To answer to the reviewer, we decided to elaborate our critic differently:

“In the context of ALS, CLR01 is able to decrease SOD1 aggregation in vitro and in vivo in the SOD1 mouse model [218,219]. However, this strategy did not succeed in slowing down motor symptoms. In contrast, CLR01 inhibiting Tau aggregation was able to ameliorate muscle-strength deterioration, anxiety-, and disinhibition-like behavior in a Alzheimer disease mouse model [220]. To our knowledge, CLR01 has not been tested in another ALS model than SOD1. CLR01 action in SOD1 is mainly explained by binding to Lysine residues, and results vary depending on the type of mutation [218]. Since CLR01 also act through Arginine binding, and that  FUS contains different arginine rich regions, it’s then tempting to speculate that CLR01 impact on FUS-ALS model might have a different outcome than the one observed in SOD1-ALS model. Neither was CLR01 tested in the C9ORF72 model to test its efficiency on PR (Proline-Arginine) or GR (glycine-Arginine) dipeptide repeat aggregation. “

-   Line 793: UBQLN2 and OPTN are not among most common mutations in fALS.

Thank you for this comment. We changed it in the manuscript, accordingly.

"Overall, there are more than 30 genes known to cause fALS. The most common mutations affect C9orf72, SOD1, FUS and TDP-43, differing in both clinical manifestation as well as treatment response [229]”

- It is not positive to cite reviews in a review (e.g. lines 605, 642, 784-785, 821-822)

Thank you for this comment. In fact, we reanalyzed the literature we referred to and discussed for every single review whether citing it was necessary. Nevertheless, in many cases the reviews we refer to compared available literature and studies and we refer to this exact comparison rather than the single studies themselves; so information that was added by the authors of those reviews and that can not be found in those single studies or anywhere else.

In detail, for the reviews that were criticized:

former line 605:

“To conclude the part on small molecules, there are many more clinical trials currently ongoing, either in phase II or I, and not detailed in here but nicely described in other recent reviews [former citation 11,107] [actual citations 11, 15, 124].”

Here we refer to reviews that describe all recent clinical trials (from phase I to III) between 2020 and 2023 and give a clear and structured overview of them. The alternative would be to cite all the different studies going on, which would exceed the limit and become very confusing. We state in the text that we are referring to reviews. As those reviews add valuable clear overviews to our manuscript – information not to be found in all involved single studies - and have differing main topics and focusses), we decided to keep them, referring solely to the mentioned therapies.

Former line 642:

Thank you for this comment. As this sentence only refers to another review without adding to our manuscript, we deleted this sentence.

Former lines 784/785 and 821/822:

This review is a result of the 2nd ALS Young Investigators Academy. The authors of the cited review and our group aim to write two complementary reviews on ALS diagnostic and therapy. The plan was - together with one of the Cells journal editors - to publish those two reviews in back-to-back with complementary information, referring to each other in order to avoid repetition in the journal. For this reason we did not remove this citation from the manuscript.

-line 856-857: Comprehensive genetic testing may not be a feasible and economical solution. Emphasis should be on finding early biomarkers.

We agree that the term “comprehensive” might be misleading. Though access to genetic testing differs from country to country, we still think that genetic testing of ALS patients at least for the common ALS genes is crucial, above all for the treatment options (e. g. Tofersen for SOD1); Knowing the causative gene also enables us to predict the clinical course (e. g. C9orf72 mutation are more prone to develop ALS/FTD -> patients may be screened for FTD in the latter case to recognize this as early as possible; disease duration which is of vast importance to the patient and decisions regarding the remaining life time, recurrence risk for other family members).

Regarding early carrier identification, both early biomarkers and genetics are crucial and should ideally go hand in hand:

If the causative mutation is known in a patient with familial ALS, other relatives at risk can undergo predictive testing, if they wish to. This is an easy and reliable method to identify pre- or asymptomatic carriers long before clinical symptoms develop. Biomarkers can then be used to assess disease conversion and to assess a good time for treatment start.

In patients with sporadic ALS, the favorable method to identify persons at risk early on are, in fact, early biomarkers. Genetic testing as screening method in healthy individuals is, currently and in our opinion, not feasible. Still, once ALS symptoms appear in sALS, genetic testing should be performed to evaluate therapeutic options and to identify other, pre- or asymptomatic mutation carriers in  relatives, as explained above.

We tried to clarify this in our manuscript and rewrote the yellow part:

“The combination of one or several of the stratification methods (genetic, molecular and clinical) will personalize the treatment of individuals and, potentially, optimize treatment response. Considering genetic testing for common genetic causes of ALS appears to be one of the first steps towards this, let alone for gene-specific treatment options such as Tofersen. An early treatment start will presumably be crucial for most future therapies, resulting in the urge for patient identification as early as possible. Apart from the development of highly sensitive and specific biomarkers pre-/asymptomatic carrier identification might be identified by offering genetic testing to family members at risk in familial/genetic ALS. Until a satisfactory causative treatment is found, supportive therapy remains of utmost importance during disease accompaniment. In this field, more studies are necessary in order to objectify relevant effects.”

Reviewer 3 Report

This is potentially useful paper for the field but it needs to be re-organized. In present form the authors do not have a clear cut way to lead the Reader through its content. Figures 1 and 2 seem not to be in line of the text flow. For example, figures should be totally synchronized with the text. All the expressions included in the circle (Figure 1) should in fact dictate the order of subchapters. At present - it is dissynchronized with each other. So the order and expressions in the Figure should dictate subtitles and and an order of subchapters. Furthermore the mechanisms of the medicines used is omitted in the first part of the review and all of sudden appears in the other subchapters (eg. Excitotoxicity). This should be inherent when the medicines have been first introduced.

The same concerns figure 2. Figure 2 has been quite chaotic since it tries to amalgamate etiological factors with experimental treatment attempts. The same concerns the comments made in the text - they are disparate to that which appears in the figure. Again Figure must dictate the order of the text!! For example - you have inflammation, neuroinflammation and something else - which are in fact synonims. What you propose in the Figure must be reflected in the text. For example etiological agents of the disease provoke the action of experimental approaches which should then follow. Otherwise we would be in a total chaos. The other way of putting it forward is to formulate Figure 3 with main experimental targets - for example - small chemical molecules (listing them in a Figure), stem cell therapy (which is of multifactorial nature), gene therapy - and so on. However, in present form Figure 2 does not dictate the text or subchapter titles - it goes separately. Therefore the Reader feeel to be lost

Author Response

This is potentially useful paper for the field but it needs to be re-organized. In present form the authors do not have a clear cut way to lead the Reader through its content. Figures 1 and 2 seem not to be in line of the text flow. For example, figures should be totally synchronized with the text.

  • All the expressions included in the circle (Figure 1) should in fact dictate the order of subchapters. At present - it is dissynchronized with each other. So the order and expressions in the Figure should dictate subtitles and an order of subchapters.

We are thankful for this suggestion and worked on synchronization of figure and text:

We changed the order of some parts as well as added headlines to allow for better structure and accordance with the figure.

Furthermore, we specified the scope of the figure focusing on supportive therapies only.

2.2. Supportive therapy
2.2.1. Mobility
2.2.1.1. Physical and occupational therapy
2.2.1.2. Robotics
2.2.2. Speech and swallowing
2.2.2.1. Speech Therapy and communication devices
2.2.2.2 Nutritional therapy
2.2.5. Ventilation

"Figure 1. Schematic overview of supportive therapy for patients with ALS: Different aspects are highlighted: a) Symptomatic medical treatment, b) Counseling and palliative care, c) Mobility, d) Speech and swallowing and e) Ventilation."

Furthermore the mechanisms of the medicines used is omitted in the first part of the review and all of sudden appears in the other subchapters (eg. Excitotoxicity). This should be inherent when the medicines have been first introduced.

The same concerns figure 2. Figure 2 has been quite chaotic since it tries to amalgamate etiological factors with experimental treatment attempts. The same concerns the comments made in the text - they are disparate to that which appears in the figure. Again Figure must dictate the order of the text!! For example - you have inflammation, neuroinflammation and something else - which are in fact synonims. What you propose in the Figure must be reflected in the text. For example etiological agents of the disease provoke the action of experimental approaches which should then follow. Otherwise we would be in a total chaos. The other way of putting it forward is to formulate Figure 3 with main experimental targets - for example - small chemical molecules (listing them in a Figure), stem cell therapy (which is of multifactorial nature), gene therapy - and so on. However, in present form Figure 2 does not dictate the text or subchapter titles - it goes separately. Therefore the Reader feeel to be lost

Thank you for this useful comment. Figure 2 has been reorganized in three figures, respecting each different topics of our review, and is now organized in accordance with the text of the manuscript.

Figure 2: Small molecules in phase II/III and III clinical trials, grouped by mode of action: a) Excitotoxicity, b) Oxidative, ER and mitochondrial stress, c) Neuroinflammation, d) Autophagy and Proteostasis, e) Neuronal death, muscle denervation and weakness and f) Bioenergetic metabolism.

Figure 3: Gene, antibody, and stem cell therapeutic approaches: a) Antisense oligonucleotides, b) Viral vectors delivering RNA interference or CRISPER/Cas9, c) Monoclonal antibody and d) Stem cell transplantation.

Figure 4: Future strategies and approaches: a) Post translational modification, b) Axonal transport, c) Nucleocytoplasmic transport, d) DNA damage and e) Molecular tweezer.

Reviewer 4 Report

In this manuscript, the authors comprehensively reviewed the current options for ALS therapy and possible future directions for development of new treatments. In addition to the description in the main text, the authors also nicely summarized this topic with two figures and one table.  However, some questions need to be addressed before getting it published in Cells.

1.      Although the paper focuses on ALS therapy, the pathogenesis of ALS needs to be briefly described, at least in the introduction section. This is important to let readers understand the principles and the targets of different kind of treatment.

2.      Line 36-38, the paper described that “Currently, three drugs with an effect on disease progression are approved”. However, based on ALS Association, There are currently six drugs approved by the U.S. Food and Drug Administration (FDA) to treat ALS and its symptoms: RELYVRIO, Radicava, Rilutek, Tiglutik, Exservan and Nuedexta.This needs to be clarified in the manuscript.

3.      Typos and formatting issues, such as:
Line 36, Too spaces before “differing”.
Line 37, “The” should be “the”.
Subtitle 3.4.4 is missing,
The format of subtitle 3.4.3 is different from others.

Author Response

In this manuscript, the authors comprehensively reviewed the current options for ALS therapy and possible future directions for development of new treatments. In addition to the description in the main text, the authors also nicely summarized this topic with two figures and one table.  However, some questions need to be addressed before getting it published in Cells.

  1. Although the paper focuses on ALS therapy, the pathogenesis of ALS needs to be briefly described, at least in the introduction section. This is important to let readers understand the principles and the targets of different kind of treatment.

In order to answer to the reviewer’s comment regarding a new introduction section on pathogenesis of ALS, we expanded the introduction that now includes the following paragraph:

“Oxidative stress, excitotoxicity, mitochondrial and proteasomal dysfunctions, RNA metabolism, altered synaptic function, disturbed axonal transport and neuroinflammation are, among other pathways, considered as important contributors to ALS disease development [10,11]. However, oxidative stress and excitotoxicity are the only two pathways targeted by the FDA approved drugs. Therefore, targeting other disease mechanisms may be key to the identification of new therapeutic drugs for additive or more effective treatment of ALS. “

  1. Line 36-38, the paper described that “Currently, three drugs with an effect on disease progression are approved”. However, based on ALS Association, “There are currently six drugs approved by the U.S. Food and Drug Administration (FDA) to treat ALS and its symptoms: RELYVRIO, Radicava, Rilutek, Tiglutik, Exservan and Nuedexta.This needs to be clarified in the manuscript.

Thank you very much for this helpful comment on our potentially misleading information. We clarified that it is in fact three different pharmacological compounds approved for ALS treatment (available as different forms and thus different drugs) and one compound approved for symptomatic treatment. We were asked to avoid brand names by the editors in our review and did only list the type of drug but not list all six brands for this reason. In the introduction we also added the information on dextromethorphan hydrobromide and quinidine sulfate (Brand not cited: Nuedextra) that is discussed later in the manuscript:

“Currently, three pharmaceutical compounds with an effect on disease progression are approved differing by country: the glutamate antagonist Riluzole (orally available in different forms: tablet, film, or liquid), the antioxidant Edaravone and the recently introduced Sodium phenylbutyrate/Taurursodiol [3–5]. By slowing down disease progression (measured by the revised ALS Functional Rating Scale, ALSFRS-R) [6], these drugs can prolong autonomy and increase survival by a few months [7–9]. A fourth agent, a combination of dextromethorphan hydrobromide and quinidine sulfate, is approved for symptomatic treatment of frontal disinhibition.”

  1. Typos and formatting issues, such as:

Line 36, Too spaces before “differing”. Line 37, “The” should be “the”.

We have revised the manuscript with a special focus on formatting issues.

Subtitle 3.4.4 is missing, The format of subtitle 3.4.3 is different from others.

Thank you for this comment, the issues were addressed.

Reviewer 5 Report

This is a very nice review of the current state of therapy in ALS. It is comprehensive and has a good future perspectives section.  It appears that the finishing of this manuscript has been rushed though and from section 3.4.1 the text is full of spelling and grammatical errors. These are far to numerous to detail but this and all subsequent sections need carefully reading and editing before resubmission.

Also, you mention in the ASO section for C9orf72 that targeting this gene may be detrimental as haploinsufficiency is one of the causes of disease. Would this not be the same for AAV miRNA targeting C9orf72? would this not be worth mentioning?

Otherwise, I enjoyed reading this review and feel it is of benefit to the wider scientific community.

Author Response

  • This is a very nice review of the current state of therapy in ALS. It is comprehensive and has a good future perspectives section.  It appears that the finishing of this manuscript has been rushed though and from section 3.4.1 the text is full of spelling and grammatical errors. These are far to numerous to detail but this and all subsequent sections need carefully reading and editing before resubmission.

We have revised the manuscript with a special focus on spelling correction and editing.

  • Also, you mention in the ASO section for C9orf72 that targeting this gene may be detrimental as haploinsufficiency is one of the causes of disease. Would this not be the same for AAV miRNA targeting C9orf72? would this not be worth mentioning?

We agree with the reviewer that AAV miRNA targeting C9orf72 might have the same detrimental outcome than using ASO against C9orf72 and therefore added your comment at the end of the paragraph and included new information on the topic :

“UniQure is currently acquiring the rights to develop the technique for Phase I/II clinical trials, which is expected for the second half of 2023. Meanwhile, the company is also developing similar techniques to treat ALS patients with C9orf72 mutations using AMT-161 miRNA molecules. However, not much is known about this future product and, regarding the negative and detrimental result obtained with ASO treatment against C9orf72, we might expect an identical outcome when targeting global C9orf72 production. In 2022, at the 29th European Society of Gene and Cell Therapy (ESGCT) Congress, UniQure also presented AAV-miQURE® results in ALS C9orf72 mouse model [166]. This strategy selectively binds the mutant repeat expansion C9orf72 mRNA, leading to degradation of the toxic target transcripts without affecting native C9orf72 mRNA and protein. Up to now, this technique looks like a much more promising therapy. “

Otherwise, I enjoyed reading this review and feel it is of benefit to the wider scientific community.

Round 2

Reviewer 1 Report

The manuscript is acceptable after the revision.

Reviewer 2 Report

I find the response of authors comprehensive and sufficient to justify publication.

Reviewer 4 Report

I would like to thank the authors for addressing my questions. The revised manuscript significantly improved.